computer modelling and simulation/ mathematical modelling/complexity

crypto-currency, critical transition, stochastic transition, critical slowing down

**Authors for correspondence:**
Chengyi Tu
e-mail: chengyitu@berkeley.edu
Samir Suweis
e-mail: samir.suweis@pd.infn.it

# Critical slowing down associated with critical transition and risk of collapse in crypto-currency

Chengyi Tu[1,2], Paolo D'Odorico[2] and Samir Suweis[3]

[1]School of Ecology and Environmental Science, Yunnan University, Kunming 650091, People's Republic of China
[2]Department of Environmental Science, Policy, and Management, University of California, Berkeley, CA 94720, USA
[3]Department of Physics and Astronomy, University of Padova, Padova 35131, Italy

CT, 0000-0001-5408-9336; SS, 0000-0002-1603-8375

The year 2017 saw the rise and fall of the crypto-currency market, followed by high variability in the price of all crypto-currencies. In this work, we study the abrupt transition in crypto-currency residuals, which is associated with the critical transition (the phenomenon of critical slowing down) or the stochastic transition phenomena. We find that, regardless of the specific crypto-currency or rolling window size, the autocorrelation always fluctuates around a high value, while the standard deviation increases monotonically. Therefore, while the autocorrelation does not display the signals of critical slowing down, the standard deviation can be used to anticipate critical or stochastic transitions. In particular, we have detected two sudden jumps in the standard deviation, in the second quarter of 2017 and at the beginning of 2018, which could have served as the early warning signals of two major price collapses that have happened in the following periods. We finally propose a mean-field phenomenological model for the price of crypto-currency to show how the use of the standard deviation of the residuals is a better leading indicator of the collapse in price than the time-series' autocorrelation. Our findings represent a first step towards a better diagnostic of the risk of critical transition in the price and/or volume of crypto-currencies.

## 1. Introduction

A crypto-currency is a digital asset/cash designed to work as a medium of exchange that uses cryptography to secure its transactions, control the creation of additional units and verify the transfer [1,2]. The first and most familiar crypto-currency to the public is Bitcoin (BTC) [3], which was launched in 2009 by an

individual/group known under the pseudonym Satoshi Nakamoto [4]. Now, over 100 000 merchants and vendors accept BTC as payment and there are 2.9 to 5.8 million unique users with a BTC wallet [5]. The success has spawned a number of competing crypto-currencies, such as Ethereum (ETH), Ripple (XPR), Litecoin (LTC), Stellar (XLM) and NEM (XEM) [1,6,7]. All crypto-currencies adopt a similar technology, based on blockchain, public ledger and reward mechanism, but they typically live on isolated transaction networks.

There is a growing body of scholarly literature on crypto-currency dynamics and their growth mechanisms. ElBahrawy *et al*. [8] showed that the dynamics of the crypto-currency volumes can be described using frameworks borrowed from ecology, such as the so-called Neutral Theory of species diversity [9–11]. These authors found that the number of species in the crypto ecosystem is stationary and some properties emerging from crypto-currency dynamics are similar to those of ecosystems. These results illustrate that the random drift and the creation at random times of new crypto-currencies (speciation) may underlie the emergence of neutral conditions. Similarly Bovet *et al*. [12] found that the abundance of certain triadic motifs in the network may be used as an early warning signal of topological collapse of the BTC's transaction network. Wu *et al*. [13] empirically verified that the market capitalizations of coins and tokens in the crypto-currency follow power law distributions with significantly different values. These authors adopted the simple stochastic proportional growth model by Malevergne *et al*. [14] to successfully recover these tail exponents. Wheatley *et al*. [15] combined a generalized Metcalfe's law with the log-periodic power law singularity model to develop a rich diagnostic of bubbles and their crashes that have punctuated the crypto-currency's history. Kondor *et al*. [16] used principal component analysis to show how structural changes accompany significant changes in the market price of BTC.

Here we focus on the dynamics of a few crypto species that cover most (more than the 85%[1]) of the crypto market cap and analyse critical transitions in crypto-currency prices. Indeed, early warning signals of critical transitions have been extensively developed to study species extinction, or sudden shifts in their populations [17–21]. One of the commonly used indicators is based on the critical slowing down (CSD) phenomenon [17,19,22,23]. If the system's dynamics approach a bifurcation point (or 'tipping point'), the dominant eigenvalue characterizing the rates of return to equilibrium after a 'small' displacement tends to zero. CSD is associated with an increase in the lag-1 autocorrelation (AR1) and—in systems driven by additive noise—in the standard deviation (Std) of the fluctuations. When the underlying nonlinear dynamics of complex systems are not known, this theory can be used to detect the proximity of a system to a critical point [24]. Several empirical studies have demonstrated how this method can be successfully applied to a variety of dynamical systems, including climatic transitions [24,25], financial markets [26,27], ecosystems [20,21,28] and brain epileptic seizures [29]. In this paper, we consider the daily close price of crypto-currencies and investigate their susceptibility to critical transitions and evaluate the risk of collapse. Research from complex systems suggests that *residuals metrics* can be used to compute generic early warning signals that may indicate for a wide class of systems if a critical threshold is approaching. In particular, it has been observed that fluctuations in population density increased in size and duration near the tipping point [19–21], in agreement with the theory. Recognizing the analogy existing between the crypto-currency market and complex ecological systems, we use the rising variability in crypto-currency prices (in our case measured by residuals) as a warning signal of systemic risk. Finally inspired by ElBahrawy *et al*. [8] and their results on emergent neutrality, we develop a mean-field phenomenological model for neutral evolution of the price of crypto-currency (instead of its market share [8]).

The paper is organized as follows: §2 shows the complete analysis of the time series of different crypto-currencies from Jan 1st 2016 to Mar 31th 2018. We focus on the time series of residuals instead of the price. We adopt AR1 and Std as the indicators of critical slowing down to predict the critical transitions and potential risk of collapse of crypto-currency price. In §3, we propose a neutral model for crypto-currency price evolution, and study numerically the effectiveness of CSD as an early warning signal for critical transitions. Discussions and conclusions are presented in §4.

## 2. Result

The year 2017 showed the rise and fall of the crypto-currency market, with a super-exponential price increase in the last quarter of the year, and a very steep downfall in December 2017. During the first quarter of 2018 the price of all crypto-currencies was highly volatile. For crypto ecosystems, we have

---

[1]Taken from https://coinmarketcap.com/charts/ corresponding to the market cap on 16 January 2017.

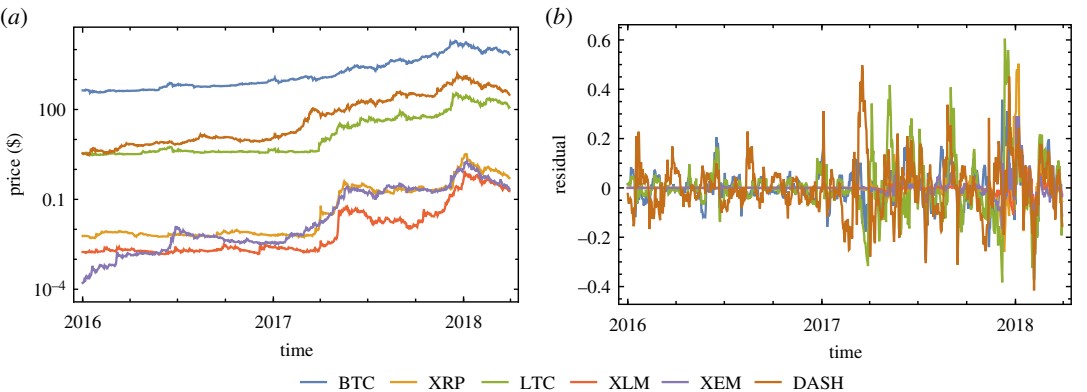

**Figure 1.** (a) Evolution of the prices of different crypto-currencies using the log scale. (b) Evolution of the residuals of different crypto-currencies using the linear scale.

**Table 1.** The properties of the residual time series of each crypto-currency, including mean, standard deviation, $p$-value of ADF test and $p$-value of KPSS test.

| crypto-currency | mean | standard deviation | ADF | KPSS |
|---|---|---|---|---|
| BTC | $1.111 \times 10^{-3}$ | $7.623 \times 10^{-2}$ | $\leq 1.000 \times 10^{-3}$ | $6.667 \times 10^{-2}$ |
| XRP | $5.492 \times 10^{-4}$ | $5.723 \times 10^{-2}$ | $\leq 1.000 \times 10^{-3}$ | $7.149 \times 10^{-2}$ |
| LTC | $1.443 \times 10^{-3}$ | $1.115 \times 10^{-1}$ | $\leq 1.000 \times 10^{-3}$ | $5.228 \times 10^{-2}$ |
| XLM | $2.494 \times 10^{-4}$ | $2.229 \times 10^{-2}$ | $\leq 1.000 \times 10^{-3}$ | $7.878 \times 10^{-2}$ |
| XEM | $3.372 \times 10^{-4}$ | $3.415 \times 10^{-2}$ | $\leq 1.000 \times 10^{-3}$ | $2.620 \times 10^{-2}$ |
| DASH | $1.694 \times 10^{-3}$ | $1.074 \times 10^{-1}$ | $\leq 1.000 \times 10^{-3}$ | $5.604 \times 10^{-2}$ |

the availability of long-term data (time series) of the crypto population abundance, and here we test if CSD could have been used to detect the presumable critical transition that took place at the end of 2017.

## 2.1. Market description and time series of residuals

Our analysis focuses on the evolution of prices (figure 1a) and residuals (figure 1b) of crypto-currencies. The dataset includes six crypto-currencies that were active in the period between Jan 1st 2016 and Mar 31st 2018 (820 days/points), namely, Bitcoin (BTC), Ripple (XPR), Litecoin (LTC), Stellar (XLM), NEM (XEM) and DASH (DASH). Because these time series clearly show trends, extreme values and non-stationarities, we log-transformed, applied filtered-detrending and then analysed the residuals from the detrending process rather than directly studying the crypto-currency prices (see Methods). Before calculating the leading indicators, the time series need to be tested for stationarity. To this end, several tests can be used. Here we combine the results of the augmented Dickey–Fuller (ADF) test [30] and the Kwiatkowski–Phillips–Schmidt–Shin (KPSS) test [31]. The null hypothesis of the ADF test is described in the Methods section. If its $p$-value is larger than the significance level, the null hypothesis is accepted (i.e. the time series is not stationary). Unlike the ADF test, the KPSS test is actually a stationarity test, i.e. its null hypothesis is that the given time series is stationary. If its $p$-value is larger than the significance level, the null hypothesis is accepted (i.e. the time series is stationary). Therefore, a time series is considered to be stationary if its ADF test is rejected and KPSS test is accepted. We find that the residuals time series are all stationary in a statistically significant way (table 1).

## 2.2. Leading indicators

To characterize the leading indicators of these crypto-currencies, we calculate the AR1 and the Std of the residuals time series (see §2.1) with respect to both large (half the size of residuals time series [17], 410 days) and small (60 days) rolling windows. Figures 2 and 3 show a summary of the results of our

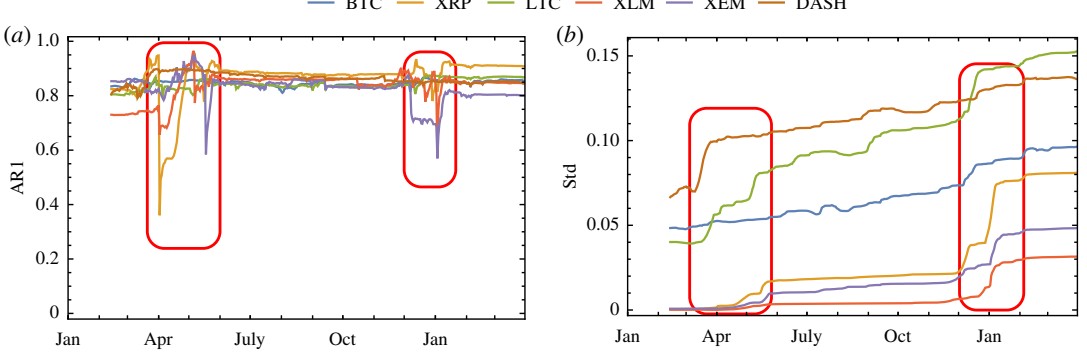

**Figure 2.** Evolution of (*a*) AR1 and (*b*) Std with respect to the large rolling window. For each curve, the value on the *x*-axis represents the last day of the calculated time period with a rolling window of 410 days.

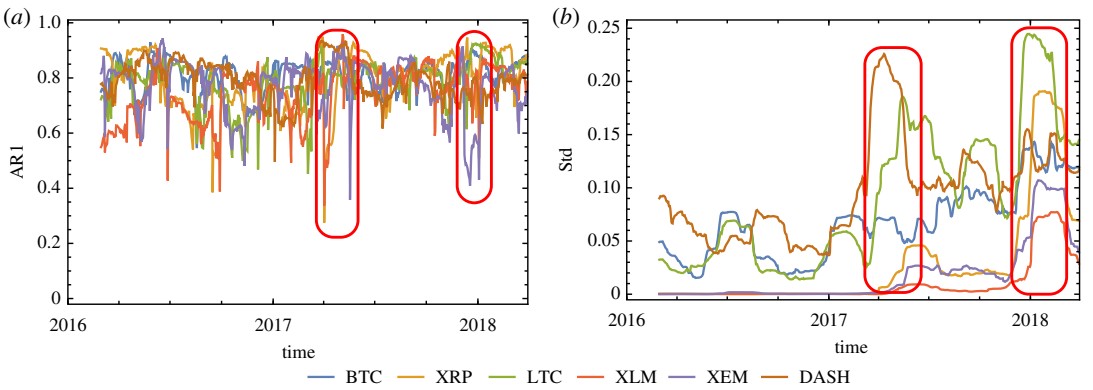

**Figure 3.** Evolution of (*a*) AR1 and (*b*) Std with respect to the small rolling window. For each curve, the value on the *x*-axis represents the last day of the calculated time period with a rolling window of 60 days.

analysis. The large rolling window analysis provides useful information on general trends, while the small rolling window analysis is more detailed and can be used to detect sudden changes.

## 2.3. CSD for large rolling window

The AR1 of the residual time series of each crypto-currency displays high values (around 0.8) during the entire period we have analysed (figure 2*a*). Therefore, it does not so effectively anticipate abrupt transitions. Nevertheless, we can detect two extreme shocks at the second quarter of 2017 and the beginning of 2018 (highlighted by the red rounded rectangles), especially for XRP and XEM. Our analysis also shows an almost monotonical increase in the Std of each crypto-currency (figure 2*b*), thus indicating the possible occurrence of CSD. In particular, we confirm the existence of the two sudden jumps in the Std time series (highlighted by the red rounded rectangles) in the second quarter of 2017, and at the beginning of 2018, exactly at the same time when abrupt changes in AR1 are observed.

## 2.4 CSD for small rolling window

The time series analysis using small rolling windows confirms the trend observed with the large rolling window (figure 3). Even though the AR1 of each crypto-currency exhibits even stronger fluctuations than in the previous case, the two main deviations denoted by the red rounded rectangles are still present. Similarly, the Std also shows strong fluctuations, but an overall increasing trend and the two sudden jumps previously observed. Using smaller rolling windows reveals more details can be seen on the fluctuations of the price residuals. In fact, despite the strong fluctuations in 2016 AR1 displays a weak statistically significant decreasing trend, which becomes positive in 2017, when it starts to increase. For XRP, XLM and XEM the first major shifts in AR1 have occurred during the second quarter of 2017. On the other hand, looking at the Std, we can also detect two sudden changes. If one considers absolute change of value, the first major jumps are observed for LTC and DASH. The crypto-currencies undergoing

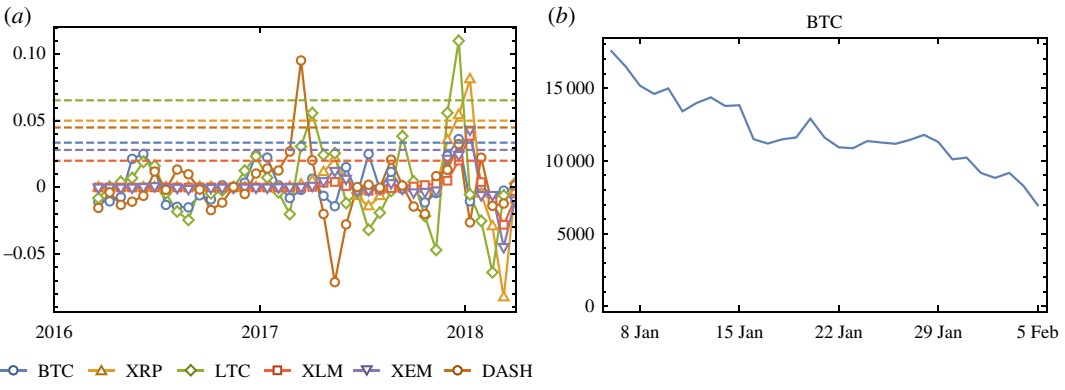

**Figure 4.** (a) $\Delta\sigma_t$ time series (the marked full line) and related threshold $\theta$ (the dashed line) for each crypto-currency. We classify an early warning signal if $\Delta\sigma_t > \theta$. (b) The price collapse event after early warning signal for BTC.

**Table 2.** Events detected using the criteria $\Delta\sigma_t > \theta$. The first column is the number of events, the second column is the crypto-currency of event, the third column is the duration of early warning signal and the fourth column is the price collapse event following the corresponding signal.

| number | crypto-currency | early warning signal | price collapse event |
|--------|-----------------|----------------------|----------------------|
| 1 | BTC | 20th Dec 2017–8th Jan 2018 | 6th Jan 2018–5th Feb 2018, decline 152% |
| 2 | XRP | 20th Dec 2017–28th Jan 2018 | 4th Jan 2018–7th Feb 2018, decline 342% |
| 3 | LTC | 20th Dec 2017–8th Feb 2018 | 18th Dec 2017–5th Feb 2018, decline 186% |
| 4 | XLM | 20th Dec 2017–28th Jan 2018 | 3rd Jan 2018–18th Feb 2018, decline 99% |
| 5 | XEM | 9th Jan 2018–28th Jan 2018 | 7th Jan 2018–30th Mar 2018, decline 733% |
| 6 | DASH | 15th Mar 2017–3rd Apr 2017 | 18th Mar 2017–11th Apr 2017, decline 80% |

the largest relative changes are XRP, XEM and XLM. While BTC never displays strong jumps, starting from 2017 it shows a stable increase in Std.

## 2.5 Criteria to evaluate early warning signals

When filtering on short-time windows, the signal is noisy. However, in this case, we can detect local trends that cannot be detected with larger rolling windows. Nevertheless, in both cases, we can detect two very strong fluctuations that can serve as an early warning signal. To better study these fluctuations, we introduce a time-dependent Std, defined as $\sigma_t$ and calculated using smaller temporal windows (see Methods section for details).

If a system approaches a critical point, its return rates back to a stable state will slow down and thus $\sigma_t$ will increase prior to the transition. We indeed find that the increase of $\sigma_t$ is an effective early warning signal of price collapse. In particular, we quantify possible early warning signals if the Std over consecutive time windows increases more than a given threshold, i.e. if $\Delta\sigma_t > \theta$ (see Methods for details on how to calculate $\Delta\sigma_t$).

Figure 4a shows $\Delta\sigma_t$ times series and related threshold $\theta$ for each crypto-currency. We look for the price collapse event (figure 1a) after each early warning signal. Figure 4b shows the price collapse event after early warning signal for BTC as an example. Table 2 shows each early warning signal and the price collapse event following the corresponding signal. We acknowledge that the proposed technique may give false positive or true negative errors that are difficult to detect. In the discussion section, we will comment on some of the events listed in table 2.

## 3. Neutral model for the evolution of crypto-currency price

ElBaharawy *et al.* [8] showed that from an ecological perspective, despite its simplicity and the assumption of no selective advantage among crypto-currencies, the so-called neutral model of

evolution is able to reproduce a number of key empirical observations (see fig. 4 in ElBaharawy *et al.* [8]). Here, we propose a simple model for the crypto-currency price (instead of its market share). In particular, we develop a mean-field phenomenological model for neutral evolution [32] of prices, introducing density-dependent fitness through Allee effects [33,34], a key ecological process observed in many systems, that disfavours rare species with respect to abundant ones. In this context, the Allee effect would disfavour crypto-currencies with low market cap, as less appealing for the buyers.

The mean-field equations of the proposed neutral model read as

$$\frac{\mathrm{d}u}{\mathrm{d}t} = (-m + r \cdot u - u^3), \tag{3.1}$$

where $u$ is the price of a given crypto-currency, the parameter $m$ represents the migration rate and $r$ the growth rate. The deterministic dynamics of this model has two stable points. The bifurcation diagram is obtained by finding the equilibria ($u^*$ at which $f(u^*) = 0$; equilibria are stable (unstable) if $\mathrm{d}f/\mathrm{d}u\,|_{u=u^*}$ less than 0 (greater than 0)). The state variable $u$ can be in one of the two stable equilibria, which correspond to higher and lower price.

We include stochasticity through a multiplicative noise term in equation (3.1) and the resulting equation is given by

$$\mathrm{d}u = (-m + r \cdot u - u^3)\,\mathrm{d}t + \sqrt{D}u\mathrm{d}W, \tag{3.2}$$

where $D$ is a parameter expressing the noise strength and $W$ is the standard uncorrelated Wiener process with zero mean value. By varying the migration rate parameter or the noise intensity, a transition between the two states may abruptly occur, and in this case, we have a critical transition from high values (bullish phase) to the lower values (bearish phase). To derive early warning signals of critical transitions, we investigate the dynamics of small deviations from the stable equilibrium $u^*$ by linearizing $f(u) = -m + r \cdot u - u^3$ around $u^*$. Analytically we cannot derive early warning signals for the residuals times series of equation (3.2), but we study them numerically.

## 3.1. Critical and stochastic transitions of mean-field phenomenological model

Our empirical analysis of early warning signals of critical transitions in crypto-currency dynamics provides evidence for increasing trends in $\sigma_t$ near the transitions with no (or statistically weak) trends in AR1. These results represent an anomaly with respect to the theory of critical transitions where AR1 is expected to increase when a system approaches the tipping point if the background noise is additive [23]. We therefore use a neutral model driven by multiplicative noise (equation (3.2)) to analyse the critical transitions and explain why $\sigma_t$ is more effective than AR1 as a leading indicator of abrupt transitions. The reason is that abrupt transitions can also occur far away from the tipping point when the strength of stochastic perturbations is large, while the theory linking critical transitions to CSD is based on the analysis of a system's response to perturbations of infinitesimal magnitude. These transitions are known as 'stochastic transition' [35,36]. By studying abrupt transitions in our neutral model under increasing strength of the noise intensity (i.e. the parameter $D$), we found that AR1 remains unaffected, while Std increases as the system approaches a (stochastic) transition. In other words, $\sigma_t$ is a good early warning signal for both stochastic transitions and critical transitions, while AR1 only works for the latter case. Figure 5 show a diagram of early warning signals of critical transitions versus stochastic transitions. Therefore, our model suggests that transitions observed in the crypto market price are stochastic transitions rather than critical ones (figure 5).

## 4. Discussion

Overall, independently of the length of the rolling windows, both CSD indicators suggest that the risk for the analysed crypto-currency to undergo price transition started to increase in early 2017, and that in particular two major shifts have occurred during the second quarter of 2017 and at the beginning of 2018 for almost all of the major crypto-currencies (in particular XEM). Indeed, a look at changes in the crypto-currency market shows that the second quarter of 2017 has been a first period of exponential growth for price and market capitalization. For instance, LTC increased 10 times in price (from 6.5\$ to 60\$) and 10 times in market capitalization (from $3 \times 10^8$ \$ to $3 \times 10^9$ \$) from the end of March 2017 to the end of August 2017; a similar pattern can be found in DASH (from 70\$ to 360\$ in price and from $5 \times 10^8$ \$ to $2.5 \times 10^9$ \$ in market capitalization) [37]. More impressively during the end of 2017, there

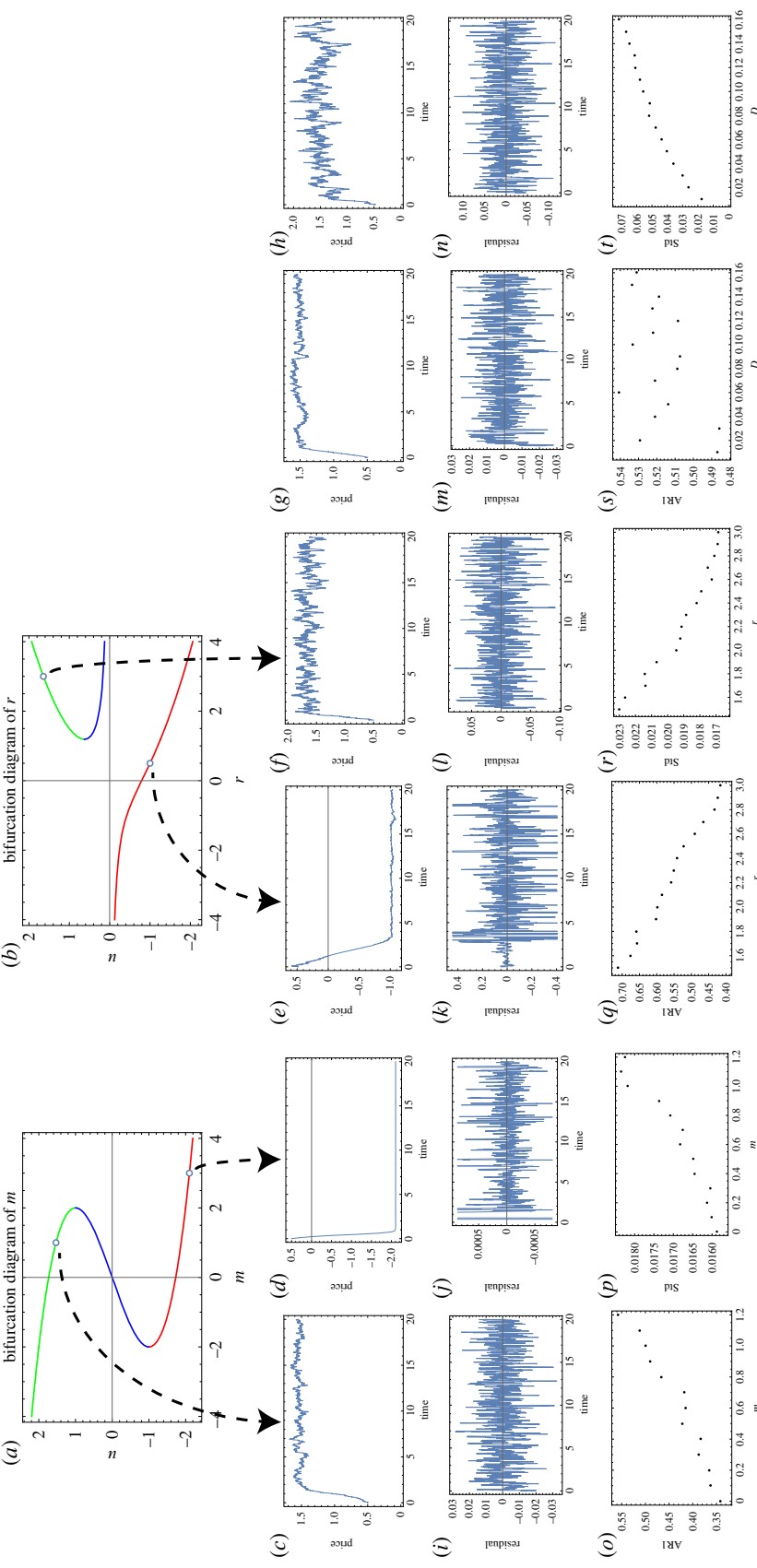

**Figure 5.** Early warning signals of critical transitions versus stochastic transitions. (*a*) The bifurcation diagram for the model with fixed $m = 0.5$, $D = 0.01$ and varying $r = [-4, 4]$. The green line represents the desired 'good' stable equilibrium, the blue line represents the unstable equilibrium and the red line represents the undesired stable equilibrium. (*c–h*) The crypto-currency price time series generated by the model with given parameter configuration: (*c*) $r = 3$, $m = 1$, $D = 0.01$; (*d*) $r = 3$, $m = 3$, $D = 0.01$; (*e*) $r = 0.5$, $m = 0.5$, $D = 0.01$; (*f*) $r = 3$, $m = 0.5$, $D = 0.01$; (*g*) $r = 3$, $m = 1$, $D = 0.01$; (*h*) $r = 3$, $m = 1$, $D = 0.16$. (*i–n*) The related residual time series obtained from the price time series (*c–h*). (*o–t*) The trend of AR1 and Std as a function of the control parameters (*m*, *r* and *D*, respectively). Each point represents the mean of 100 realizations. Panels (*o,p*) indicate the existence of critical slowing down phenomenon for the 'endogenous' critical transition controlled by the parameter m. Panels (*q,r*) highlight how 'intrinsic' critical transitions controlled by the growth rate *r* can be anticipated by Std and AR1. Panels (*s,t*) show how Std is a good indicator for the phenomenon of stochastic transition (controlled by the parameter D), while AR1 is not.

has been a steep exponential increase in price of all major crypto-currencies, followed by a $\approx$40% loss during the first month of 2018 [37]. In particular at the end of January 2018, XEM was affected by the biggest hack in the history of the technology (approximate 533 million $ in XEM stolen), with important consequences for the XEM price drop in the following weeks (it lost about 80% of its value in few weeks). All these factors are reflected in very large fluctuations in the AR1 and in big jumps in the Std. An interesting result is that in March 2018, the Std rapidly decreased to the value it had before the occurrence of the shock at the beginning of 2018. This observed reduction in price fluctuations preceded a steady increase in BTC price (a $\approx$30% increase in April). Because BTC is the leading crypto-currency with the largest market capitalization, the price behaviours of other crypto-currencies are somehow driven by BTC. We will quantitatively tackle this problem in a future work.

In this paper, we have investigated the CSD indicators of six representative crypto-currencies between Jan 1st 2016 and Mar 31st 2018. The gradual change of some underlying conditions may lead to a system closer to a critical point causing a loss of resilience, with smaller perturbations being able to induce a shift to the alternative state [17,19,22,23]. CSD indicates that the system is approaching this critical point, and thus its return time to equilibrium upon a small system perturbation strongly increases. The consequent increase in AR1 and Std is often used as an early warning of the critical transition. Even when the underlying dynamics of the complex system are unknown or limited time series are available, the early warning signatures still exist, and leading indicators can be used to detect critical points before they shift the state. A large number of studies have now demonstrated the potential application of these leading indicators as warning signals of increased risk for upcoming state transitions. But increasing AR1 or/and Std cannot guarantee that the system approaches the critical point [38,39]. In general, the loss of the system's tendency to return to its equilibrium may cause it to simply undergo stochastic perturbations. For all crypto-currencies, the AR1 was found to fluctuate around high values without displaying trends consistent with CSD. On the other hand, Std exhibited a tendency to increase throughout the study period independently of the size of the rolling window. These results have been confirmed also using a phenomenological model of neutral evolution for the crypto-currency price. Early warning signals of a collapse in these crypto-currencies can be found in two sudden steps in Std in the second quarter of 2017 and the beginning of 2018. These results suggest the price transitions observed in our analysis are due to stochastic rather than critical transitions. Anyway, predicting when the price collapse will strike remains a highly challenging problem and our findings represent only a first step in the direction of improving the understanding and prediction of the risk of the crypto-currency collapse.

# 5. Methods

## 5.1. Data

The daily (close) prices of crypto-currencies are downloaded from the Website Coin Market Cap [37]. We just consider the most important crypto-currencies that were active in the period from 1st Jan 2016 to 31st Mar 2018. These crypto-currencies include Bitcoin (BTC), Ripple (XPR), Litecoin (LTC), Stellar (XLM) and NEM (XEM) and DASH (DASH).

## 5.2. Log-transform

There are some zero and extreme values in the price time series of each crypto-currency, so some preprocessing is warranted. Here, we log-transform it by $x_t = \log (z_t - 1)$ where $z_t$ is the price time series of a crypto-currency at time $t$, with $t = 1, \ldots , N \in \{N\}$, i.e. $N$ is the length of time series. In words, $x_t$ is the price time series after log-transform. This step does not influence the sensitivity of early warning signals because it does not change the distribution of the original data [17].

## 5.3. Filtering-detrending

We adopt a Gaussian filter to remove trends and filter out the high frequencies [17,24]. Detrending was motivated by the fact that time series with strong trends or periodicities tends to exhibit a strong correlation structure, which would affect the use of AR1 as a leading indicator of critical transitions. Likewise, high-frequency fluctuations can cause spurious indications of impending transitions. When applying the filter, care was taken to not overfit or filter out the slow dynamics embedded in the time series. Therefore, we set the radius of Gaussian filter equal to 30 days.

## 5.4. Stationarity test

The prerequisite of CSD is that the time series is stationary. A time series is said to be weakly stationary, if its mean and autocovariance do not vary with respect to time, i.e. $E[x_t] = \mu < \infty$ and $E[(x_t - \mu)(x_{t+h} - \mu)] = \gamma(h)$ for all $t$, $h \in \{N\}$ and $h > 0$. We apply the ADF test [30] and KPSS test [31] to check whether the time series is stationary [40]. For a given time series, the minimum number of differences required in order to obtain a weakly stationary series is called the order of integration and denoted by $I(d)$ where d is the minimum number of differences. Additionally, $I(1)$ time series is also called unit root. The ADF test performs a hypothesis testing on the time series with the null hypothesis that it is unit root and the alternative hypothesis is that the time series is stationary. Smaller $p$-value is associated with higher probability that the tested time series is stationary. By contrast, the KPSS test is actually a stationarity test, meaning that the null hypothesis is that the time series is stationary.

## 5.5. Critical slowing down and early warning signal for price collapse detection

Several leading indicators, such as AR1 and Std have been proposed as early warning signals to detect the proximity of a system to a critical point. An increase in AR1 indicates that the state of the system has become increasingly similar between consecutive observations and it is given by $AR1_t = E[(x_t - \mu_t)(x_{t+1} - \mu_t)]/\sigma_t^2$ where $\mu_t = 1/n \sum_{s=1}^{n} x_{t-n+s}$ is the mean of $x_t$ during the rolling time window of length $n$ (from $t - n + 1$ to $t$) and $\sigma_t = \sqrt{1/n \sum_{s=1}^{n} (x_{t-n+s} - \mu_t)^2}$ is the Std of $x_t$ during the same time rolling window (for large rolling window $n = 410$ and for small rolling window $n = 60$ and $t = n, \dots, N$). If a system approaches a critical point, its return rates back to a stable state will slow down and thus $\sigma_t$ will increase prior to the transition. We indeed find that the increase of $\sigma_t$ is an effective early warning signal of price collapse.

In particular, we define an early warning signal if the Std increases more than a given threshold, i.e. if $\Delta\sigma_t > \theta$, where $\Delta\sigma_t = \sigma_t - \sigma_{t-20}$ denotes the difference between the Stds calculated over consecutive time intervals of 20-day length and $\theta$ is a threshold. The thresholding criteria is introduced to quantitatively define and consistently recognize early warning signals of critical transitions. Of course this method is sensitive to the selected threshold (i.e. if $\theta$ is 'too low', we will detect many early warning signals, while if it is 'too high' we will detect only a few early warning signals). Although there is not any *a-priori* fully objective criteria in order to choose this threshold, a standard and often used criteria is to fix it between one or three times the standard deviation of the Std time series, i.e. $\sigma_t$ with $t = n, \dots, N$. In our case, we set the threshold to catch fluctuations greater than one Std of $\sigma_t$, i.e. $\theta = \sqrt{1/(N - n + 1) \sum_{t=n}^{N} (\sigma_t - \mu_{\sigma_t})^2}$. If $\Delta\sigma_t > \theta$ at time $t$, we define the duration of early warning sign as 20 days (from $t - 20$ to $t$). Further, if both $\Delta\sigma_t = \sigma_t - \sigma_{t-20} > \theta$ and $\Delta\sigma_{t-20} = \sigma_{t-20} - \sigma_{t-40} > \theta$ then the duration of the early warning sign is 40 days. Of course these durations depend on the choice of the time window duration (in this case 20 days) and so they are simply an indication of the persistence of the early warning signal. Then, we calculate the price collapse (figure 1a) by looking at the price time series $(z_t)$ and define it as the largest decline in price after the first early warning signal. The price collapse duration is the time interval between the first day of the price collapse and the day corresponding to the end of the price decline. The first day of the price collapse event is later than the first day of its early warning signal, but it may be earlier than the end of the early warning signal duration. Therefore, there is the possibility of an overlap between the period considered for the 'early detection' and the effective 'price collapse event' as in fact found in table 2.

## 5.6. Procedure summary

In summary, to predict a critical transition in the crypto-currency price time series, we perform the following steps:

Step 1. Log-transformation of the price time series.
Step 2. Application of a Gaussian filter to obtain the residuals time series.
Step 3. Checking the stationarity of the residuals time series. If so, go to next step.
Step 4. Setting a small and large window respectively, and then calculating the $AR1_t$ and $\sigma_t$.
Step 5. Detect early warning signals (if $\Delta\sigma_t > \theta$).

Data accessibility. The dataset used in this study is public and can be found in Coin Market Cap [37]. The web-scraped data and code can be found on Dryad Repository. The Dryad URL is https://doi.org/10.5061/dryad.t4j13fm and its DOI is https://doi.org/10.5061/dryad.t4j13fm.

Authors' contributions. S.S. and P.D. designed the study. C.T. collected, preprocessed and analysed the data. S.S. developed the neutral model, C.T. simulated the model and analysed the critical transitions. P.D. and S.S. wrote the manuscript. All authors gave final approval for publication.

Competing interests. We have no competing interests.

Funding. We received no funding for this study.

Acknowledgements. C.T. acknowledges the part of the financial support comes from Yunnan University project C176210103. S.S. acknowledges Visiting Program Cariparo grant TEAMS and BioReact STARS 2018 UNIPD grant.

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
