## [Reviewer comments · Royal Society Open Science]

Review History

RSOS-181097.R0 (Original submission)

Review form: Reviewer 1

Is the manuscript scientifically sound in its present form?

No

Are the interpretations and conclusions justified by the results?

No

Is the language acceptable?

Yes

Is it clear how to access all supporting data?

Yes

Do you have any ethical concerns with this paper?

No

Have you any concerns about statistical analyses in this paper?

Yes

Recommendation?

Major revision is needed (please make suggestions in comments)

Comments to the Author(s)

Dear Authors, I hope that you find the feedback below to be helpful.

METHODS & ARGUMENT

- Please prove that the residuals metrics are leading rather than coincidental -- or even lagging -- the price. I understand that the Gaussian smoothing used is not causal. Is the smoothing updated with each time step, or is it only done once on the full dataset hence using future information? To visualize, the window around the 2018 crash in Fig 1A and 2A should be zoomed in and compared with the price in a figure. Further, there have been many "mini" and a few not so mini bubbles in the history of Bitcoin. However, volatility of returns (and I guess filter residuals) tends to spike after rather than before. How does this reconcile with the one instance argued in the manuscript? Further, in Fig 3A the AR1 signal seems far too noisy to provide any predictive information. Not to mention that no threshold or hazard function for regime changes is specified.

- In figure 1, as the authors suggest, 2017 was a year of high volatility. This is reflected in the residuals (as well as in the returns -- not shown). Mixing these two low and high volatility regimes as done in Fig 2B is probably mostly the cause of the increasing trending of residual standard deviation. This concern is largely confirmed by Fig 3A. In terms of the jumps in the residual standard deviation, it seems that there are plenty of jumps in the short history of Bitcoin and other cryptos which are not predictive of a tipping point. Hence Fig 1B seems not to be predictive. Please resolve.

- These metrics rely heavily on the validity of the detrending by filter, as the Std and AR1 measures largely just measure the size and persistence of deviations from the filter line. How can you justify this filtering as consistently recovering the trend, where it is often thought that bubbles form super-exponential trajectories? E.g., the jumps in autocorrelation and standard deviation observed in Fig 2A can simply mean that there is growth that is too fast for your "30 day bandwidth" filter (unsurprising given the periods of explosive growth). Hence, the approach of [2] is more direct.

More minor points:

- Is AR(1) sufficient? How much structure is left in the residuals of the AR(1) fit?
- De-trending is easier said than done and can introduce spurious autocorrelations (see: Slutsky-Yule effect). Further, the unit root tests are far from perfect, and not very powerful, meaning that they may miss consequential non-stationarities left behind by the filter.

COMPOSITION

Intro:

- I would leave the background on cryptocurrency to an appendix, or better still, simply point to a more standard reference if available.
- The paper is a bit long on summarizing the ElBaharawy et al. findings, whose prediction of the bitcoin market share has proven to be pretty far off. You may wish to mention the view given in [1], which makes different assumptions.
- Further, in [2] similar authors have looked at cryptocurrency bubbles, with a more specific view on speculative bubbles as a critical phenomena, with tipping point predictability. This should be reflected.

REFERENCES

- [1] Wu, Ke, Spencer Wheatley, and Didier Sornette. "Classification of cryptocurrency coins and tokens by the dynamics of their market capitalizations." *Royal Society Open Science* 5.9 (2018): 180381.
- [2] Wheatley, S., Sornette, D., Reppen, M., Huber, T., & Gantner, R. N. (2018). Are Bitcoin Bubbles Predictable. Combining a Generalised Metcalfe's Law and the LPPLS Model. Swiss Finance Institute Research Paper, (18-22).

Review form: Reviewer 2

Is the manuscript scientifically sound in its present form?

Yes

Are the interpretations and conclusions justified by the results?

Yes

Is the language acceptable?

Yes

Is it clear how to access all supporting data?

Yes

Do you have any ethical concerns with this paper?

No

Have you any concerns about statistical analyses in this paper?

No

Recommendation?

Reject

Comments to the Author(s)

Review of "Critical slowing down associated with critical transition and risk of collapse in cryptocurrency"

The paper investigates how simple metrics, coming from a general theory of critical transitions, can serve as a early-warning signal for the risk of large changes in cryptocurrencies.

The authors apply a statistical analysis to a set of time series of prices of cryptocurrencies, aimed at identifying warning signals of catastrophic changes, as inspired by ecological literature, Three main comments arise from reading this paper on cryptocurrencies:

- the authors don't make any theoretical contribution to the theory of critical slowing down for critical transitions detection,
- the authors don't make specific efforts in understanding why the small set of cryptocurrencies studied, which represent a very specific context for testing such general theory, should be particularly relevant or informative for the general theory, and how their results quantitatively relate to results in other fields, e.g. ecology,
- the authors appear to identify in total two individual cases, which might be considered as "true positive" predictions of the theory in this context, although they don't specify a quantitative binary criterion for predicting large changes in a near future, so it is not clear at this stage whether these two cases are actually predicted by their theory and whether the theory would predict also other large changes that did not occur.

Overall, the authors carefully apply an existing methodology to a specific dataset of cryptocurrencies, provide a statistical analysis of the time series of prices, and find a partial indication of a possibly interesting direction of research in the field of cryptocurrencies.

Based on these observations, I would suggest the authors to revise and resubmit the paper to a journal more specifically interested in the quantitative study of the cryptocurrencies market.

Review form: Reviewer 3

Is the manuscript scientifically sound in its present form?

No

Are the interpretations and conclusions justified by the results?

No

Is the language acceptable?

Yes

Is it clear how to access all supporting data?

Yes

Do you have any ethical concerns with this paper?

No

Have you any concerns about statistical analyses in this paper?

No

Recommendation?

Reject

Comments to the Author(s)

The authors aim at anticipating the cryptocurrency market crisis by analyzing the prices series of few selected currencies. Specifically, they look for evidence for the “critical slowing down” phenomenon, which occur in any continuous model approaching a fold bifurcation. Their analysis consists in measuring the autocorrelation and the standard deviation of the residuals time series over a sliding window.

The problem of anticipating the cryptocurrency market crisis is extremely interesting within the literature and relevant for a variety of stakeholders. The approach adopted is potentially interesting but the design of the article and the analysis is extremely limited in the current state.

First, the authors provide no theoretical framework justifying the choice to adopt the “critical slowing down” phenomenon approach. Critical slowing down is observed as a system approaches a fold bifurcation point, but, from the current article, it is unclear why and when exactly the cryptocurrency market would undergo such a transition. Literature in finance has suggested that critical slowing down indicators are weak to predict to global financial crisis, so it is not evident that critical transitions in financial markets are preceded by critical slowing down. The authors should definitely provide a section reviewing literature in finance and a more solid theoretical background.

Then, I find the scope of the analysis unclear and the tools used very limited. The authors should provide a clear description of what they define as a “critical transitions” in the cryptocurrency market, and how long before a transition their method allows to anticipate it. The current method is qualitative, since the detection of periods where the standard deviation and autocorrelation are

supposedly “increasing” is not automatized. The analysis focuses only on few selected currencies, and the role played by the temporal sampling of data is not discussed.

Finally, there is little care for the figures design. Figure 2A and 3A (which are central figures in the paper) are extremely hard to interpret. In Figures 2A and 2B the x-axis should only include the period 2017-2018.

For these reasons, I judge this article does not provide a useful contribution to the literature.

Decision letter (RSOS-181097.R0)

04-Jan-2019

Dear Dr Tu:

Manuscript ID RSOS-181097 entitled "Critical slowing down associated with critical transition and risk of collapse in cryptocurrency" which you submitted to Royal Society Open Science, has been reviewed. The comments from reviewers are included at the bottom of this letter.

In view of the criticisms of the reviewers, the manuscript has been rejected in its current form. However, a new manuscript may be submitted which takes into consideration these comments.

Please note that resubmitting your manuscript does not guarantee eventual acceptance, and that your resubmission will be subject to peer review before a decision is made.

Your resubmitted manuscript should be submitted by 04-Jul-2019. If you are unable to submit by this date please contact the Editorial Office.

Please note that Royal Society Open Science will introduce article processing charges for all new submissions received from 1 January 2018. Charges will also apply to papers transferred to Royal Society Open Science from other Royal Society Publishing journals, as well as papers submitted as part of our collaboration with the Royal Society of Chemistry (<http://rsos.royalsocietypublishing.org/chemistry>). If your manuscript is submitted and accepted for publication after 1 Jan 2018, you will be asked to pay the article processing charge, unless you request a waiver and this is approved by Royal Society Publishing. You can find out more about the charges at <http://rsos.royalsocietypublishing.org/page/charges>. Should you have any queries, please contact openscience@royalsociety.org.

on behalf of Professor Martina Sasse (Associate Editor) and Marta Kwiatkowska (Subject Editor)
 openscience@royalsociety.org

Associate Editor Comments to Author (Professor Martina Sasse):

Associate Editor: 1

Comments to the Author:

Three reviewers have assessed your paper. The broad view is that it is not yet ready for publication. However, as each reviewer provides extensive feedback for improvement, the journal is willing to consider a substantially revised manuscript that fully addresses the concerns raised by the referees. If, for whatever reason, you cannot incorporate their requested changes, you should provide a fully reasoned scientific rebuttal for excluding the change. If you resubmit, the manuscript will be returned to at least these reviewers for their advice. If they remain unsatisfied by your changes, we will not be able to consider any further revisions unless there are exceptional reasons for doing so. Good luck and we look forward to receiving the resubmission.

Reviewers' Comments to Author:

Reviewer: 1

Comments to the Author(s)

Dear Authors, I hope that you find the feedback below to be helpful.

METHODS & ARGUMENT

- Please prove that the residuals metrics are leading rather than coincidental -- or even lagging -- the price. I understand that the Gaussian smoothing used is not causal. Is the smoothing updated with each time step, or is it only done once on the full dataset hence using future information? To visualize, the window around the 2018 crash in Fig 1A and 2A should be zoomed in and compared with the price in a figure. Further, there have been many "mini" and a few not so mini bubbles in the history of Bitcoin. However, volatility of returns (and I guess filter residuals) tends to spike after rather than before. How does this reconcile with the one instance argued in the manuscript? Further, in Fig 3A the AR1 signal seems far too noisy to provide any predictive information. Not to mention that no threshold or hazard function for regime changes is specified.

- In figure 1, as the authors suggest, 2017 was a year of high volatility. This is reflected in the residuals (as well as in the returns -- not shown). Mixing these two low and high volatility regimes as done in Fig 2B is probably mostly the cause of the increasing trending of residual standard deviation. This concern is largely confirmed by Fig 3A. In terms of the jumps in the residual standard deviation, it seems that there are plenty of jumps in the short history of Bitcoin and other cryptos which are not predictive of a tipping point. Hence Fig 1B seems not to be predictive. Please resolve.

- These metrics rely heavily on the validity of the detrending by filter, as the Std and AR1 measures largely just measure the size and persistence of deviations from the filter line. How can you justify this filtering as consistently recovering the trend, where it is often thought that bubbles form super-exponential trajectories? E.g., the jumps in autocorrelation and standard deviation observed in Fig 2A can simply mean that there is growth that is too fast for your "30 day bandwidth" filter (unsurprising given the periods of explosive growth). Hence, the approach of [2] is more direct.

More minor points:

- Is AR(1) sufficient? How much structure is left in the residuals of the AR(1) fit?

- De-trending is easier said than done and can introduce spurious autocorrelations (see: Slutsky-Yule effect). Further, the unit root tests are far from perfect, and not very powerful, meaning that they may miss consequential non-stationarities left behind by the filter.

COMPOSITION

Intro:

- I would leave the background on cryptocurrency to an appendix, or better still, simply point to a more standard reference if available.
- The paper is a bit long on summarizing the ElBaharawy et al. findings, whose prediction of the bitcoin market share has proven to be pretty far off. You may wish to mention the view given in [1], which makes different assumptions.
- Further, in [2] similar authors have looked at cryptocurrency bubbles, with a more specific view on speculative bubbles as a critical phenomena, with tipping point predictability. This should be reflected.

REFERENCES

- [1] Wu, Ke, Spencer Wheatley, and Didier Sornette. "Classification of cryptocurrency coins and tokens by the dynamics of their market capitalizations." *Royal Society Open Science* 5.9 (2018): 180381.
- [2] Wheatley, S., Sornette, D., Reppen, M., Huber, T., & Gantner, R. N. (2018). Are Bitcoin Bubbles Predictable. Combining a Generalised Metcalfe's Law and the LPPLS Model. Swiss Finance Institute Research Paper, (18-22).

Reviewer: 2

Comments to the Author(s)

Review of "Critical slowing down associated with critical transition and risk of collapse in cryptocurrency"

The paper investigates how simple metrics, coming from a general theory of critical transitions, can serve as a early-warning signal for the risk of large changes in cryptocurrencies.

The authors apply a statistical analysis to a set of time series of prices of cryptocurrencies, aimed at identifying warning signals of catastrophic changes, as inspired by ecological literature, Three main comments arise from reading this paper on cryptocurrencies:

- the authors don't make any theoretical contribution to the theory of critical slowing down for critical transitions detection,
- the authors don't make specific efforts in understanding why the small set of cryptocurrencies studied, which represent a very specific context for testing such general theory, should be particularly relevant or informative for the general theory, and how their results quantitatively relate to results in other fields, e.g. ecology,
- the authors appear to identify in total two individual cases, which might be considered as "true positive" predictions of the theory in this context, although they don't specify a quantitative binary criterion for predicting large changes in a near future, so it is not clear at this stage whether these two cases are actually predicted by their theory and whether the theory would predict also other large changes that did not occur.

Overall, the authors carefully apply an existing methodology to a specific dataset of cryptocurrencies, provide a statistical analysis of the time series of prices, and find a partial indication of a possibly interesting direction of research in the field of cryptocurrencies.

Based on these observations, I would suggest the authors to revise and resubmit the paper to a journal more specifically interested in the quantitative study of the cryptocurrencies market.

Reviewer: 3

Comments to the Author(s)

The authors aim at anticipating the cryptocurrency market crisis by analyzing the prices series of few selected currencies. Specifically, they look for evidence for the "critical slowing down" phenomenon, which occur in any continuous model approaching a fold bifurcation. Their analysis consists in measuring the autocorrelation and the standard deviation of the residuals time series over a sliding window.

The problem of anticipating the cryptocurrency market crisis is extremely interesting within the literature and relevant for a variety of stakeholders. The approach adopted is potentially interesting but the design of the article and the analysis is extremely limited in the current state.

First, the authors provide no theoretical framework justifying the choice to adopt the “critical slowing down” phenomenon approach. Critical slowing down is observed as a system approaches a fold bifurcation point, but, from the current article, it is unclear why and when exactly the cryptocurrency market would undergo such a transition. Literature in finance has suggested that critical slowing down indicators are weak to predict to global financial crisis, so it is not evident that critical transitions in financial markets are preceded by critical slowing down. The authors should definitely provide a section reviewing literature in finance and a more solid theoretical background.

Then, I find the scope of the analysis unclear and the tools used very limited. The authors should provide a clear description of what they define as a “critical transitions” in the cryptocurrency market, and how long before a transition their method allows to anticipate it. The current method is qualitative, since the detection of periods where the standard deviation and autocorrelation are supposedly “increasing” is not automatized. The analysis focuses only on few selected currencies, and the role played by the temporal sampling of data is not discussed.

Finally, there is little care for the figures design. Figure 2A and 3A (which are central figures in the paper) are extremely hard to interpret. In Figures 2A and 2B the x-axis should only include the period 2017-2018.

For these reasons, I judge this article does not provide a useful contribution to the literature.

Author's Response to Decision Letter for (RSOS-181097.R0)

See Appendix A.

RSOS-191450.R0

Review form: Reviewer 3

Is the manuscript scientifically sound in its present form?

No

Are the interpretations and conclusions justified by the results?

No

Is the language acceptable?

Yes

Do you have any ethical concerns with this paper?

No

Have you any concerns about statistical analyses in this paper?

Yes

Recommendation?

Major revision is needed (please make suggestions in comments)

Comments to the Author(s)

The authors present a methodology to anticipate collapses in cryptocurrency prices. The method is based on the theory of “Critical Slowing Down” phenomenon, which suggests that tipping points are preceded by an increase in autocorrelation and standard deviation of price fluctuations.

The authors find no evidence for increase in autocorrelation, but they find some examples in which collapses of cryptocurrency prices were preceded by an increase of standard deviation.

Anticipating cryptocurrency prices and their market behaviour is of paramount importance for stakeholder and investors in the cryptocurrency market. The article is clearly written and suitable for the broad readership of Royal Society Open Science. However, I have some concerns regarding the methodology that should be addressed for the article to be ready for publication. Namely, I find the author should implement a clear and well-described procedure to identify “price collapse” events. Then, they should evaluate the ability of their presented methodology in identifying such events. The way the article is written now (results of these procedure are presented based on few successful examples), there is no way to evaluate if the methodology is meaningful or not. Also, there is an overlap between the period considered for the “early detection” and the effective “price collapse event” (see table 2). This should be resolved to make sure the method is effective.

Below, I present some more detailed comments:

Page 3, line 7. “These results illustrate that the random drift and the creation at random times of new crypto-currencies (speciation) underlie the emergence of neutral conditions.” Please add “may underlie”.

Page 3, line 22. Quantify “most”.

Page 4, line 29. “Method” → “Methods”

Page 8, line 1. Please clarify the difference between what you defined as σ and what you defined as σ .

Table 2. Please define more clearly how you identified “events”. There is an overlap between the periods corresponding to the warning signal and the event. Are you sure that the early warning signal detected effectively occurs before the drop in price?

Page 9, line 27. Compilation error.

Review form: Reviewer 4

Is the manuscript scientifically sound in its present form?

Yes

Are the interpretations and conclusions justified by the results?

Yes

Is the language acceptable?

Yes

Do you have any ethical concerns with this paper?

No

Have you any concerns about statistical analyses in this paper?

No

Recommendation?

Accept as is

Comments to the Author(s)

The paper presents an interesting analysis of cryptocurrency volatility, drawing parallels with ecological systems. The claim is that by utilizing rigorous statistical analysis, one extracts the amount of volatility by computing the standard deviation, autocorrelation coefficients, and other statistics in the cryptocurrency pricing data, and uses it to predict downturns in the broader market. The authors use daily pricing data on several major cryptocurrencies, including Bitcoin, in their analysis. The pricing data is run through preprocessing before analysis which seems consistent with typical data analysis. The conclusion is that the resulting autocorrelation coefficients exhibit a period of increased volatility before a downturn in prices.

Decision letter (RSOS-191450.R0)

12-Dec-2019

Dear Dr Tu,

The Subject Editor assigned to your paper ("Critical slowing down associated with critical transition and risk of collapse in cryptocurrency") has now received comments from reviewers. We would like you to revise your paper in accordance with the referee and Associate Editor suggestions which can be found below (not including confidential reports to the Editor). Please note this decision does not guarantee eventual acceptance.

Please submit a copy of your revised paper before 04-Jan-2020. Please note that the revision deadline will expire at 00.00am on this date. If we do not hear from you within this time then it will be assumed that the paper has been withdrawn. In exceptional circumstances, extensions may be possible if agreed with the Editorial Office in advance. We do not allow multiple rounds of revision so we urge you to make every effort to fully address all of the comments at this stage. If deemed necessary by the Editors, your manuscript will be sent back to one or more of the original reviewers for assessment. If the original reviewers are not available we may invite new reviewers.

When submitting your revised manuscript, you must respond to the comments made by the referees and upload a file "Response to Referees" in "Section 6 - File Upload". Please use this to document how you have responded to each of the comments, and the adjustments you have made. In order to expedite the processing of the revised manuscript, please be as specific as possible in your response.

- Ethics statement

- Data accessibility

<http://datadryad.org/submit?journalID=RSOS&manu=RSOS-191450>

- Competing interests

- Authors' contributions

- Acknowledgements

- Funding statement

Kind regards,
Anita Kristiansen

Editorial Coordinator
 Royal Society Open Science
 openscience@royalsociety.org

on behalf of Marta Kwiatkowska (Subject Editor)
 openscience@royalsociety.org

Subject Editor Comments to Author (Marta Kwiatkowska):

One of the reviewers suggests improvements to the paper regarding strengthening of the methodology.

Reviewer comments to Author:

Reviewer: 3

Comments to the Author(s)

The authors present a methodology to anticipate collapses in cryptocurrency prices. The method is based on the theory of “Critical Slowing Down” phenomenon, which suggests that tipping points are preceded by an increase in autocorrelation and standard deviation of price fluctuations.

The authors find no evidence for increase in autocorrelation, but they find some examples in which collapses of cryptocurrency prices were preceded by an increase of standard deviation.

Anticipating cryptocurrency prices and their market behaviour is of paramount importance for stakeholder and investors in the cryptocurrency market. The article is clearly written and suitable for the broad readership of Royal Society Open Science. However, I have some concerns regarding the methodology that should be addressed for the article to be ready for publication. Namely, I find the author should implement a clear and well-described procedure to identify “price collapse” events. Then, they should evaluate the ability of their presented methodology in identifying such events. The way the article is written now (results of these procedure are presented based on few successful examples), there is no way to evaluate if the methodology is meaningful or not. Also, there is an overlap between the period considered for the “early detection” and the effective “price collapse event” (see table 2). This should be resolved to make sure the method is effective.

Below, I present some more detailed comments:

Page 3, line 7. “These results illustrate that the random drift and the creation at random times of new crypto-currencies (speciation) underlie the emergence of neutral conditions.” Please add “may underlie”.

Page 3, line 22. Quantify “most”.

Page 4, line 29. “Method” → “Methods”

Page 8, line 1. Please clarify the difference between what you defined as std and what you defined as σ .

Table 2. Please define more clearly how you identified “events”. There is an overlap between the periods corresponding to the warning signal and the event. Are you sure that the early warning signal detected effectively occurs before the drop in price?

Page 9, line 27. Compilation error.

Reviewer: 4

Comments to the Author(s)

The paper presents an interesting analysis of cryptocurrency volatility, drawing parallels with ecological systems. The claim is that by utilizing rigorous statistical analysis, one extracts the amount of volatility by computing the standard deviation, autocorrelation coefficients, and other

statistics in the cryptocurrency pricing data, and uses it to predict downturns in the broader market. The authors use daily pricing data on several major cryptocurrencies, including Bitcoin, in their analysis. The pricing data is run through preprocessing before analysis which seems consistent with typical data analysis. The conclusion is that the resulting autocorrelation coefficients exhibit a period of increased volatility before a downturn in prices.

Author's Response to Decision Letter for (RSOS-191450.R0)

See Appendix B.

RSOS-191450.R2 (Revision)

Review form: Reviewer 3

Is the manuscript scientifically sound in its present form?

Yes

Are the interpretations and conclusions justified by the results?

Yes

Is the language acceptable?

Yes

Do you have any ethical concerns with this paper?

No

Have you any concerns about statistical analyses in this paper?

No

Recommendation?

Accept as is

Comments to the Author(s)

My comments have been addressed.

Decision letter (RSOS-191450.R1)

27-Feb-2020

Dear Dr Tu,

It is a pleasure to accept your manuscript entitled "Critical slowing down associated with critical transition and risk of collapse in cryptocurrency" in its current form for publication in Royal Society Open Science. The comments of the reviewer(s) who reviewed your manuscript are included at the foot of this letter.

Please ensure that you send to the editorial office an editable version of your accepted

manuscript, and individual files for each figure and table included in your manuscript. You can send these in a zip folder if more convenient. Failure to provide these files may delay the processing of your proof. You may disregard this request if you have already provided these files to the editorial office.

on behalf of Marta Kwiatkowska (Subject Editor)
openscience@royalsociety.org

Reviewer comments to Author:
Reviewer: 3

Comments to the Author(s)
My comments have been addressed.

Appendix A

Reply to the Reviewers' Comments:

Reviewer: 1

First of all, we want to thank the reviewer 1 for her/his careful reading of our work and very relevant and helpful comments. We have considered all the reviewer suggestions, making the related changes in the manuscript. When we have not incorporated the requested changes, we have provided reasoned and referenced explanations, and we have anyway rewritten the related parts in the manuscript to make them clearer. We feel that thank to reviewer 1, now our manuscript is greatly improved. We report reviewer's comments in *italic*.

- Please prove that the residuals metrics are leading rather than coincidental -- or even lagging -- the price.
Research from complex systems suggest that *residuals metrics* can be used to compute generic early-warning signals that may indicate for a wide class of systems if a critical threshold is approaching¹. In particular, it has been observed that fluctuations of population density increased in size and duration near the tipping point, in agreement with the theory². Our work starts from the analogy of crypto-currency market as complex ecological systems and as also suggested in studies of early warning on financial market rising variability (in our case measured by residuals) could signal systemic risk³. We have added a paragraph to clarify this point.

I understand that the Gaussian smoothing used is not causal. Is the smoothing updated with each time step, or is it only done once on the full dataset hence using future information?

The Gaussian smoothing/filter is updated at each time step. In fact, we adopted Gaussian filter with radius 30 meaning that it uses the backward and forward local windows of 15 time steps. This is a standard filtering procedure also in other studies of cryptocurrency market⁴⁻⁶.

To visualize, the window around the 2018 crash in Fig 1A and 2A should be zoomed in and compared with the price in a figure.

Considering the Reviewer's suggestion, we have re-plotted Fig 2. The new panels C and D zoom in the crash windows and compare with the price.

Volatility of returns (and I guess filter residuals) tends to spike after rather than before. How does this reconcile with the one instance argued in the manuscript?

A key difference between variance as we have computed and the measure of volatility is as follows. We computed variance of the residuals, which is obtained by removing longer time scale trends of the stock market time series. Volatility, on the other hand, is a measure of variance in the rate of return calculated from price of stocks without removing any time trends in the data. They both are measures of variability but as the reviewer suggest volatility of returns tends to spike after rather than before, while several studies show that variance of residuals may be used as early warning of critical transition^{3,5-9}.

In Fig 3A the AR1 signal seems far too noisy to provide any predictive information.

As the reviewer pointed out, when filtering on short time windows the signal is noisy. However, in this case we can detect local trends that cannot be detected with larger rolling windows. The latter analysis in fact provides useful information only on general trends. Nevertheless, in both cases we can detect two very strong fluctuations that can serve as an early warning signal.

To quantify possible early warning signs, we introduced the threshold θ , so that conditions in which $|\Delta Std| > \theta$, where Δ denotes the difference between consecutive time intervals of 20-day-length are considered as early warning signals of a transition. We focus on the Std of the residuals as possible early warning signal because the previous analysis shows that, unlike AR1, Std exhibits signals of CSD.

These thresholding criteria are introduced to quantitatively define and consistently recognize early warning signals of tipping points. Of course this method is highly sensitive to the selected threshold (i.e., if θ is "too low", we will detect many early warnings, while if it is "too high" we will detect only few events. However, setting θ in a reasonable range with respect to data means selecting θ between one or three standard deviation of the residuals time series. In our case we have set the threshold to catch fluctuations greater than one

standard deviation, $\theta = \sigma$. Each crypto-currency thus has a different threshold θ , depending on the standard deviation of the residual times series. We list all these warning event in the following table.

Events detected using the criteria $|\Delta Std| > \theta$. The first column is the number of events, the second column is the crypto-currency of event, the third column is the duration of early warning signal and the fourth column is the collapse event following the corresponding signal.

Number	Cryptocurrency	Duration	Event
1	BTC	20th Dec 2017 - 8th Jan 2018	6th Jan 2018 - 5th Feb 2018, decline 152%
2	XRP	20th Dec 2017 - 28th Jan 2018	4th Jan 2018 - 7th Feb 2018, decline 342%
3	LTC	20th Dec 2017 - 8th Feb 2018	18th Dec 2017 - 5th Feb 2018, decline 186%
4	XLM	20th Dec 2017 - 28th Jan 2018	3th Jan 2018 - 18th Feb 2018, decline 99%
5	XEM	9th Jan 2018 - 28th Jan 2018	7th Jan 2018 - 30th Mar 2018, decline 733%
6	DASH	15th Mar 2017 - 3th Apr 2017	18th Mar 2017 - 11st Apr 2017, decline 80%

[...]No threshold or hazard function for regime changes is specified.

In this paper, we investigate the Critical Slowing Down (CSD) phenomenon as early warning signals of critical transitions. If the system's dynamics approach a bifurcation point (or 'tipping point'), its lag-1 autocorrelation (AR1) and standard deviation (Std) will increase. In many studies^{3,5,6,9}, this approach – differently from typical use of the threshold or hazard functions to detect regime changes – does not use a quantitative threshold to define the early warning event. Nevertheless, as explained above, we think that the suggestion of the reviewers is very pertinent, and thus we decided to add this new analysis where we indeed introduce a threshold for the definition of the early warning signals.

There are plenty of jumps in the short history of Bitcoin and other cryptos which are not predictive of a tipping point.

Filtering should help to remove these irrelevant small jumps, and by introducing the threshold criteria we now have an "objective" criterion to define early warning events for tipping points. Of course, the proposed technique may give false positive or true negative errors. In Table 2 and in the revised text we comment these limitations.

How can you justify this filtering as consistently recovering the trend, where it is often thought that bubbles form super-exponential trajectories?

We thank the reviewer to justify this point. As we hope now is clearer, early warning based on CSD are based on the use of AR1 and variance of residuals as leading indicators of critical transitions. Detrending is motivated by the fact that time series with strong trends or periodicities tends to exhibit a strong correlation structure, which would affect these quantities. Likewise, high-frequency fluctuations can cause spurious indications of impending transitions. Gaussian filter is a common method to remove such trends and filter out high frequencies.

Is AR(1) sufficient? How much structure is left in the residuals of the AR(1) fit?

If the system's dynamics approach a bifurcation point (or 'tipping point'), the dominant eigenvalue characterizing the rates of return to equilibrium after a 'small' displacement tends to zero. Therefore, its AR1 increases. This is true independently of how much structure is left in the residuals of the AR1. In fact, when the underlying non-linear dynamics of complex systems are not known, this theory can be used to detect the proximity of a system to a critical point^{7,10,11}.

De-trending is easier said than done and can introduce spurious autocorrelations. the unit root tests are far from perfect

In this paper, we would not like to discuss the drawback of de-trending. Gaussian filter is a common method to remove trends and filter out the high frequencies. Additionally, applying other type of detrending or filtering.

e.g. first-differences, removing running means, loess smoothing, will obtain similar results. We tested for stationarity by augmented Dickey-Fuller (ADF) test and the Kwiatkowski-Phillips-Schmidt-Shin (KPSS) test.

The ADF test performs a hypothesis testing on the time series with the null hypothesis that it is unit root and the alternative hypothesis is that the time series is stationary. Smaller p-value are associated with higher probability that the tested time series is stationary. By contrast, the KPSS test is actually a stationarity test, meaning that the null hypothesis is that the time series is stationary. A time series is considered to be stationary, if its ADF test is rejected and KPSS test is accepted.

I would leave the background on cryptocurrency to an appendix, or better still, simply point to a more standard reference if available.

We have re-written this part according to the Reviewer's suggestion.

The paper is a bit long on summarizing the ElBaharawy et al. findings, whose prediction of the bitcoin market share has proven to be pretty far off. You may wish to mention the view given in [1], which makes different assumptions.

Considering the Reviewer's suggestion, we have re-written the summary of ElBaharawy et al. In addition, we also cite this and related important references in Introduction.

Further, in [2] similar authors have looked at cryptocurrency bubbles, with a more specific view on speculative bubbles as a critical phenomena, with tipping point predictability. This should be reflected.

Thank you for your suggestion. We have cited this relevant reference.

Reviewer: 2

We want to thank the reviewer 2 for her/his comments that have helped us to better explain the scope and goals of our work, clarify and rewrite some points and add further analysis. We hope that reviewer 2 will find the revised version of our manuscript improved.

The authors don't make any theoretical contribution to the theory of critical slowing down for critical transitions detection

The focus of this work was to investigate possible early warning signs of critical transitions in the cryptocurrency market based on Critical Slowing Down (CSD) phenomenon. To our knowledge there are not previous attempt to apply CSD to this interesting case. Several empirical studies have demonstrated how this method can be successfully applied to a variety of dynamical systems (but without adding any novel theoretical methodological contributions to CSD^{2,10,12-15}). Nevertheless, we appreciate the suggestion of the reviewer about a possible more theoretical contribution to the work, and **we thus decided to add an entire section so to illustrate possible existence of early warnings also for a mean-field model of neutral dynamics displaying abrupt transitions.**

El-Baharawy et al.¹⁶ showed that from an ecological perspective, despite its simplicity and the assumption of no selective advantage among crypto-currencies, the so-called neutral model of evolution is able to reproduce a number of key empirical observations (see Fig. 4 in El-Baharawy et al.¹⁶). Here, we propose a simple model for the crypto-currency price (instead of its market share). In particular, we develop a mean-field phenomenological model for neutral evolution¹⁷ of prices, introducing density dependent fitness through Allee effects^{18,19}, a key ecological process observed in many systems, that disfavors rare species with respect to abundant one. In this context, the Allee effect would disfavour crypto-currencies with low market cap, as less appealing by the buyers.

The mean field equations of the proposed neutral model read as:

$$\frac{du}{dt} = (-m + r \cdot u - u^3) \quad (1)$$

where u is the price of a given cryptocurrency, the parameter m represents the migration rate (i.e., endogenous causes affecting the market price), r the growth rate (intrinsic dynamics of the market price). The deterministic dynamics of this model has two stable points. The bifurcation diagram is obtained by finding the equilibria (u^*

at which $f(u^*) = 0$; equilibria are stable (unstable) if $df/du|_{u=u^*} < 0$ (> 0). The state variable u can be in one of the two stable equilibria, which correspond to higher and lower price.

We include stochasticity through a multiplicative noise term in Eq. **Error! Reference source not found.** and the resulting equation is given by

$$du = (-m + r \cdot u - u^3)dt + \sqrt{D}u dW \quad (2)$$

where D is the diffusion constant, and W is the standard uncorrelated Wiener process with zero mean value. By varying the migration rate parameter or the noise intensity, a transition between the two states may abruptly occur, and in this case we have a critical transition from high values (bullish phase) to the lower values (bearish phase).

To derive early warning signals of critical transitions, the idea is to investigate the dynamics of small deviations from the stable equilibrium u^* by linearizing $f(u) = -m + r \cdot u - u^3$ around u^* . Analytically we cannot derive early warning signs for the residuals times series of Eq. (2), but we numerically study them and we show that thanks to the model we are able to explain why Std is more effective than AR1 to anticipate critical transitions.

We finally note that CSD is a general method that is not specific to the data we are analysing. The gradual change of some underlying conditions may lead to a system closer to a critical point causing a loss of resilience, with smaller perturbations being able to induce a shift to the alternative state. CSD indicates that the system is approaching this critical point, and thus its return time to equilibrium upon a small system perturbation strongly increases. Even when the underlying dynamics of the complex system are unknown or limited time series are available, the early warning signals still exist, and the leading indicators can be used to detect critical points before they shift the state. A large number of studies have now demonstrated the potential application of these leading indicators as warning signs of increased risk for upcoming state transitions.

For all these reasons, and because our mean-field model provides proof of concept showing possible critical transition detected by AR1 and Std, we have decided to use CSD as early warning of critical transition in the crypto-market. We have added all these points in the revised manuscript.

The authors don't make specific efforts in understanding why the small set of cryptocurrencies studied, which represent a very specific context for testing such general theory, should be particularly relevant or informative for the general theory, and how their results quantitatively relate to results in other fields.

We thank the reviewer to give the opportunity to clarify this point. We have used the set of cryptocurrencies that between 2016 and 2018 was covering most of the total cryptocurrency market cap. Therefore, we feel that we are considering representative data to describe general behaviour of the crypto world. Moreover, CSD is a general method that is not specific to the data we are analysing. The gradual change of some underlying conditions may lead to a system closer to a critical point causing a loss of resilience, with smaller perturbations being able to induce a shift to the alternative state. CSD indicates that the system is approaching this critical point, and thus its return time to equilibrium upon a small system perturbation strongly increases. Even when the underlying dynamics of the complex system are unknown or limited time series are available, the early warning signatures still exist, and the leading indicators can be used to detect critical points before they shift the state. A large number of studies have now demonstrated the potential application of these leading indicators as warning signals of increased risk for upcoming state transitions.

In particular, the analogy with the neutral ecological models are considering here (and proposed in^{17,20-22}), allows us to say that our results are also informative and can be related to ecological systems. In particular CSD for the mean field model for neutral of evolution with density dependent Allee effects is a novel result that can be applied also in the context of neutral theory.

The authors appear to identify in total two individual cases, which might be considered as "true positive" predictions of the theory in this context, although they don't specify a quantitative binary criterion for predicting large changes in a near future, so it is not clear at this stage whether these two cases are actually predicted by their theory and whether the theory would predict also other large changes that did not occur.

To quantify possible early warning signs, we introduced the threshold θ , so that conditions in which $|\Delta Std| > \theta$, where Δ denotes the difference between consecutive time intervals of 20-day-length are considered as early warning signals of a transition. We focus on the Std of the residuals as possible early warning signal because the previous analysis shows that, unlike AR1, Std exhibits signals of CSD.

These thresholding criteria are introduced to quantitatively define and consistently recognize early warning signals of tipping points. Of course this method is highly sensitive to the selected threshold (i.e., if θ is “too low”, we will detect many early warnings, while if it is “too high” we will detect only few events. However, setting θ in a reasonable range with respect to data means selecting θ between one or three standard deviation of the residuals time series. In our case we have set the threshold to catch fluctuations greater than one standard deviation, $\theta = \sigma$. Each crypto-currency thus has a different threshold θ , depending on the standard deviation of the residual times series.

Where possible, we try to identify and comments all the events detected by CSD and comment related limitations. We list all these warning events in the new Table 2.

Reviewer: 3

We thank reviewer 3 for careful reading of our work. The comments are in lines with the main comments of the other reviewers, and thus we feel that the reviews have strongly helped us to increase the quality our work. We hope that that the reviewer will find our revised manuscript remarkably improved.

The authors provide no theoretical framework justifying the choice to adopt the “critical slowing down” phenomenon approach.

We appreciate the suggestion of the reviewer about providing a more theoretical explanation on the choice to adopt “critical slowing down” phenomenon approach. We thus have decided to add a section so to illustrate possible existence of early warnings also for a mean-field model of neutral dynamics displaying abrupt transitions.

El-Baharawy et al.¹⁶ showed that from an ecological perspective, despite its simplicity and the assumption of no selective advantage among crypto-currencies, the so-called neutral model of evolution is able to reproduce a number of key empirical observations (see Fig. 4 in El-Baharawy et al.¹⁶). Here, we propose a simple model for the crypto-currency price (instead of its market share). In particular, we develop a mean-field phenomenological model for neutral evolution¹⁷ of prices, introducing density dependent fitness through Allee effects^{18,19}, a key ecological process observed in many systems, that disfavors rare species with respect to abundant one. In this context, the Allee effect would disfavour crypto-currencies with low market cap, as less appealing by the buyers.

The mean field equations of the proposed neutral model read as:

$$\frac{du}{dt} = (-m + r \cdot u - u^3) \quad (3)$$

where u is the price of a given cryptocurrency, the parameter m represents the migration rate (i.e., endogenous causes affecting the market price), r the growth rate (intrinsic dynamics of the market price). The deterministic dynamics of this model has two stable points. The bifurcation diagram is obtained by finding the equilibria (u^* at which $f(u^*) = 0$; equilibria are stable (unstable) if $df / du|_{u=u^*} < 0$ (> 0). The state variable u can be in one of the two stable equilibria, which correspond to higher and lower price.

We include stochasticity through a multiplicative noise term in Eq. **Error! Reference source not found.** and the resulting equation is given by

$$du = (-m + r \cdot u - u^3)dt + \sqrt{D}udW \quad (4)$$

where D is the diffusion constant, and W is the standard uncorrelated Wiener process with zero mean value. By varying the migration rate parameter or the noise intensity, a transition between the two states may abruptly occur, and in this case we have a critical transition from high values (bullish phase) to the lower values (bearish phase).

To derive early warning signals of critical transitions, the idea is to investigate the dynamics of small deviations from the stable equilibrium u^* by linearizing $f(u) = -m + r \cdot u - u^3$ around u^* . Analytically we cannot derive early warning signs for the residuals times series of Eq. (2), but we numerically study them and we show that thanks to the model we are able to explain why Std is more effective than AR1 to anticipate critical transitions.

We finally note that CSD is a general method that is not specific to the data we are analysing. The gradual change of some underlying conditions may lead to a system closer to a critical point causing a loss of resilience, with smaller perturbations being able to induce a shift to the alternative state. CSD indicates that the system is approaching this critical point, and thus its return time to equilibrium upon a small system perturbation strongly increases. Even when the underlying dynamics of the complex system are unknown or limited time series are available, the early warning signatures still exist, and the leading indicators can be used to detect critical points before they shift the state. A large number of studies have now demonstrated the potential application of these leading indicators as warning signs of increased risk for upcoming state transitions.

For all these reasons, and because our mean field model provides proof of concept showing possible critical transition detected by AR1 and Std, we have decided to use CSD as early warning of critical transition in the crypto-market. We have added all these points in the revised manuscript.

The authors should definitely provide a section reviewing literature in finance and a more solid theoretical background.

Considering the Reviewer's suggestion, we have re-written the introduction section also providing a more solid background on *literature in finance* and theoretical foundations on CSD and neutral theory.

The authors should provide a clear description of what they define as a "critical transitions" in the cryptocurrency market, and how long before a transition their method allows to anticipate it.

To quantify possible early warning signs, we introduced the threshold θ , so that conditions in which $|\Delta Std| > \theta$, where Δ denotes the difference between consecutive time intervals of 20-day-length are considered as early warning signals of a transition. We focus on the Std of the residuals as possible early warning signal because the previous analysis shows that, unlike AR1, Std exhibits signals of CSD.

By introducing these thresholding criteria, we now have an "objective" way to define early warning events for tipping points. Of course, the proposed technique may give false positive or true negative errors. Where possible, we try to identify and comments all the events detected by CSD and comment related limitations.

Figure 2A and 3A (which are central figures in the paper) are extremely hard to interpret. In Figures 2A and 2B the x-axis should only include the period 2017-2018.

Considering the Reviewer's suggestion, we have re-plotted Fig. 2. The x-axis of panel A and B only include the valid periods.

- 1 Scheffer, M. *et al.* Early-warning signals for critical transitions. *Nature* **461**, 53-59, doi:10.1038/nature08227 (2009).
- 2 Dai, L., Vorselen, D., Korolev, K. S. & Gore, J. J. S. Generic indicators for loss of resilience before a tipping point leading to population collapse. **336**, 1175-1177 (2012).
- 3 Guttal, V., Raghavendra, S., Goel, N. & Hoarau, Q. Lack of Critical Slowing Down Suggests that Financial Meltdowns Are Not Critical Transitions, yet Rising Variability Could Signal Systemic Risk. *PLoS One* **11**, e0144198, doi:10.1371/journal.pone.0144198 (2016).
- 4 Белінський, А. О. & Соловійов, В. М. Complex network precursors of crashes and critical events in the cryptocurrency market. (2018).
- 5 Diks, C., Hommes, C. & Wang, J. Critical slowing down as an early warning signal for financial crises? *Empirical Economics*, doi:10.1007/s00181-018-1527-3 (2018).
- 6 Gatfaoui, H., Nagot, I. & De Peretti, P. in *Systemic Risk Tomography* 73-93 (Elsevier, 2017).
- 7 Dakos, V. *et al.* Methods for detecting early warnings of critical transitions in time series illustrated using simulated ecological data. *PloS one* **7**, e41010 (2012).
- 8 Dakos, V., Van Nes, E. H., D'Odorico, P. & Scheffer, M. Robustness of variance and autocorrelation as indicators of critical slowing down. *Ecology* **93**, 264-271 (2012).
- 9 Tan, J. P. L. & Cheong, S. S. A. Critical slowing down associated with regime shifts in the US housing market. *The European Physical Journal B* **87**, doi:10.1140/epjb/e2014-41038-1 (2014).
- 10 Scheffer, M. *Critical transitions in nature and society*. (Princeton University Press, 2009).
- 11 Scheffer, M. *et al.* Early-warning signals for critical transitions. *Nature* **461**, 53 (2009).
- 12 Dakos, V. *et al.* Slowing down as an early warning signal for abrupt climate change. *Proceedings of the National Academy of Sciences* **105**, 14308-14312 (2008).
- 13 Drake, J. M. & Griffen, B. D. Early warning signals of extinction in deteriorating environments. *Nature* **467**, 456 (2010).
- 14 Carpenter, S. R. *et al.* Early warnings of regime shifts: a whole-ecosystem experiment. *Science* **332**, 1079-1082 (2011).
- 15 Veraart, A. J. *et al.* Recovery rates reflect distance to a tipping point in a living system. *Nature* **481**, 357 (2012).
- 16 ElBahrawy, A., Alessandretti, L., Kandler, A., Pastor-Satorras, R. & Baronchelli, A. Evolutionary dynamics of the cryptocurrency market. *Royal Society open science* **4**, 170623 (2017).
- 17 Azaele, S., Pigolotti, S., Banavar, J. R. & Maritan, A. Dynamical evolution of ecosystems. *Nature* **444**, 926-928, doi:10.1038/nature05320 (2006).
- 18 Courchamp, F., Berec, L. & Gascoigne, J. *Allee effects in ecology and conservation*. (Oxford University Press, 2008).
- 19 Allee, W. & Bowen, E. S. Studies in animal aggregations: mass protection against colloidal silver among goldfishes. *Journal of Experimental Zoology Part A: Ecological Genetics and Physiology* **61**, 185-207 (1932).
- 20 Volkov, I., Banavar, J. R., Hubbell, S. P. & Maritan, A. Neutral theory and relative species abundance in ecology. *Nature* **424**, 1035-1037 (2003).
- 21 Volkov, I., Banavar, J. R., He, F., Hubbell, S. P. & Maritan, A. Density dependence explains tree species abundance and diversity in tropical forests. *Nature* **438**, 658-661, doi:10.1038/nature04030 (2005).
- 22 Volkov, I., Banavar, J. R., Hubbell, S. P. & Maritan, A. Patterns of relative species abundance in rainforests and coral reefs. *Nature* **450**, 45-49, doi:10.1038/nature06197 (2007).

Appendix B

Reply to the Reviewers' Comments:

Reviewer: 3

First of all, we want to thank the reviewer 3 for her/his careful reading of our work and very relevant and helpful comments. We have considered all the reviewer suggestions, making the related changes in the manuscript. We feel that thank to reviewer 3, now our manuscript is greatly improved. We report reviewer's comments in *italic*.

(1) I find the author should implement a clear and well-described procedure to identify “price collapse” events. Then, they should evaluate the ability of their presented methodology in identifying such events. The way the article is written now (results of these procedure are presented based on few successful examples), there is no way to evaluate if the methodology is meaningful or not.

We thank the reviewer to highlight this important point. In fact we have realized that many details were not present in the methods sections, and therefore the procedure to identify “price collapse” was not clearly described. In the revised manuscript **we have added a new section titled “Critical slowing down and early warning signal for price collapse detection”**. In this section we now clearly illustrate all the steps to identify early warning of price critical transitions. In particular, our analysis is focused on the detection of systematic

increase of the (log-transformed and filtered) times series x_t , i.e. $\sigma_t = \sqrt{\frac{1}{n} \sum_{s=1}^n (x_{t-n+s} - \mu_t)^2}$, during the rolling time window of duration n (from $t - n + 1$ to t with $t = n, \dots, N$, and for small rolling window $n = 60$), where

$\mu_t = \frac{1}{n} \sum_{s=1}^n x_{t-n+s}$ is the mean of x_t . If a system approaches a critical point, its return rates back to a stable state

will slow down and thus σ_t will increase prior to the transition. We indeed find that the increase of σ_t is an effective early warning signal of price collapse. In particular, we define an early warning signal if the standard deviation increases more than a given threshold, i.e. if $\Delta\sigma_t > \theta$, where $DS_t = S_t - S_{t-20}$ denotes the difference between the Std calculated over consecutive time intervals of 20-day-length and θ is a threshold. The thresholding criteria is introduced to quantitatively define and consistently recognize early warning signals of critical transitions. Of course this method is sensitive to the selected threshold (i.e., if θ is “too low”, we will detect many early warning signals, while if it is “too high” we will detect only few early warning signals). Although there is not any *a-priori* fully objective criteria in order to choose this threshold, a standard and often used criteria is to fix it between one or three times the standard deviation of the Std time series, i.e. S_t with

$t = n, \dots, N$. In our case we set the threshold to catch fluctuations greater than one standard deviation of S_t , i.e.

$$\theta = \sqrt{\frac{1}{N-n+1} \sum_{t=n}^N (\sigma_t - \mu_{\sigma_t})^2}.$$

In summary to predict a critical transition in the crypto-currency price time series we perform the following steps:

Step 1. Log-transformation of the price time series

Step 2. Application of a Gaussian filter to obtain the residuals time series

Step 3. Checking the stationarity of the residuals time series. If so, go to next step.

Step 4. Setting a small and large window respectively, and then calculating the $AR1_t$ and σ_t .

Step 5. Detect early warning signals (if $\Delta\sigma_t > \theta$).

We list all “price collapse” event in Table 2. These events correspond to those detected “visually” in Figure 3. We acknowledge that the proposed technique may give false positive or true negative errors that are difficult to detect. In the discussion section we have commented on some of the events listed in Table 2.

(2) *There is an overlap between the period considered for the “early detection” and the effective “price collapse event” (see table 2).*

If $\Delta\sigma_t > \theta$ at time t , we define the duration of early warning signal as 20 days (from $t-20$ to t). Further, if both $DS_t = S_t - S_{t-20} > q$ and $DS_{t-20} = S_{t-20} - S_{t-40} > q$ then the duration of early warning signal is 40 days. Of course these durations depends on the choice of the time window duration (in this case 20 days) and so they are simply an indication of the persistence of the early warning signal. Then, we calculate the price collapse (Fig. 1 A) by looking at the price time series (z_t) and define it as largest decline in price after the first early warning signal. The price collapse duration is the time interval between the first day of the price collapse and the day corresponding to the end of the price decline. The first day of the price collapse event is later than the first day of its early warning signal, but it may be earlier than the end of the early warning signal duration. Therefore, there is the possibility of an overlap between the period considered for the “early detection” and the effective “price collapse event”, as observed by the reviewer. We have added this comment in the revised version.

(2) *I present some more detailed comments:*

- Page 3, line 7. *“These results illustrate that the random drift and the creation at random times of new cryptocurrencies (speciation) underlie the emergence of neutral conditions.” Please add “may underlie”.*
- Page 3, line 22. *Quantify “most”.*
- Page 4, line 29. *“Method” → “Methods”*
- Page 8, line 1. *Please clarify the difference between what you defined as std and what you defined as σ .*
- Page 9, line 27. *Compilation error*

We have made all the corrections according to the Reviewer’s comments. Thanks